# Cloud-Based Software Architecture for Fully Automated Point-of-Care Molecular Diagnostic Device [note 1]

**DOI:** 10.3390/s21216980

**Published:** 2021-10-21

**Authors:** Byeong-Heon Kil, Ji-Seong Park, Mun-Ho Ryu, Chan-Young Park, Yu-Seop Kim, Jong-Dae Kim

**Affiliations:** 1School of Software, Hallym University, Chuncheon-si 24252, Korea; zsewa0@hallym.ac.kr (B.-H.K.); cypark@hallym.ac.kr (C.-Y.P.); yskim@hallym.ac.kr (Y.-S.K.); 2Bio-IT Research Center, Hallym University, Chuncheon-si 24252, Korea; 3Biomedux Co., Ltd., Suwon-si 16226, Korea; qkrwltjd426@gmail.com; 4Division of Biomedical Engineering, Jeonbuk National University, Jeonju 54896, Korea; mhryu@jbnu.ac.kr; 5Research Center of Healthcare & Welfare Instrument for the Aged, Jeonbuk National University, Jeonju 54896, Korea

**Keywords:** fully automated molecular diagnostic system, point-of-care, cloud-based, web-based user interface

## Abstract

This paper proposes a cloud-based software architecture for fully automated point-of-care molecular diagnostic devices. The target system operates a cartridge consisting of an extraction body for DNA extraction and a PCR chip for amplification and fluorescence detection. To facilitate control and monitoring via the cloud, a socket server was employed for fundamental molecular diagnostic functions such as DNA extraction, amplification, and fluorescence detection. The user interface for experimental control and monitoring was constructed with the RESTful application programming interface, allowing access from the terminal device, edge, and cloud. Furthermore, it can also be accessed through any web-based user interface on smart computing devices such as smart phones or tablets. An emulator with the proposed software architecture was fabricated to validate successful operation.

## 1. Introduction

Infectious diseases are a major burden to global health and the economy, especially in developing countries [1,2,3,4,5]. Despite the significant efforts to enhance global health along with the advancement in health care technologies, reducing the number of deaths occurring annually in resource-limited settings remains a challenge [6]. Viruses such as Zika, Chikungunya, Dengue Fever, Malaria, HIV, Ebola, COVID-19, and other emerging pathogens are of significant threat, in particular owing to their high infectivity and lethality. To prevent global spread of such highly contagious pathogens, early diagnosis through consistent monitoring is crucial. Furthermore, a diagnostic equipment to identify specific pathogens and its strain is required [7]. Deploying point-of-care (POC) diagnostic equipment to developing countries can help overcome the economic burden that rises from scarce resources, which will enhance the medical care.

In general, viral culture, serological diagnosis, and nucleic acid detection are used for POC diagnosis [2,8,9]. Viral culture diagnosis is difficult to conduct in resource-limited settings since it requires highly skilled professional and expensive equipment with a long turnaround time for the results. Similarly, serological cultures are also restricted since it requires a complex antibody engineering step. Nucleic acid diagnosis, especially the polymerase chain reaction (PCR), is favored over the other two methods owing to the accurate and relatively rapid diagnosis time. Through PCR, the unique target sequence of each pathogen in the ribonucleic acid (RNA) or deoxyribonucleic acid (DNA) is replicated to a billion fold [10,11]. Therefore, PCR provides the most accurate results, making it the gold standard in detecting pathogenic markers. The COVID-19 pandemic outbreak further highlights the advantages of PCR, where real-time PCR (qPCR), which detects fluorescence signals during amplification, is used as the official diagnosis method [12,13]. However, commercially available qPCR equipment is yet to be portable and cost-effective, requiring professionals to transport the device. Therefore, patient samples are transported to central laboratories or hospitals, prolonging the turnaround time and restricting the ability to be used at POC in resource-limited settings [2,14,15,16,17,18,19]. In addition, POC qPCR devices can not only benefit developing countries, but also in any emergency rooms. For example, patients admitted to emergency rooms suspecting a Gram-negative bacteria infection, respiratory diseases, and tuberculosis require rapid, on-site diagnosis to determine the appropriate antibiotic and treatment [20]. In some cases of influenza, Tamiflu needs to be administered within 48 h of observing the first symptom [2,21]. Evidently, the development of portable, user-friendly, and cost-effective qPCR equipment is critical in promoting global health [16,22].

The World Health Organization (WHO) set the criteria ‘ASSURED’ for ideal POC diagnostic devices [23,24]. To meet these criteria, recent studies working on POC devices aim to develop a device that is Affordable, Sensitive, Specific, User-friendly, Rapid and robust, Equipment-free, and Deliverable (ASSURED). This is represented in the device by being cost-effective, having portability, requiring less sample volume, and producing rapid results [25,26,27]. Another factor that is studied for POC devices is the automation of the process from DNA extraction to PCR and detection, which will achieve the goal of sample-to-answer [28].

The transformation of biomedical instrument platforms to big data platforms is speculated from the example of the platform change that occurred following the expansion of the graphical user interface (GUI). The implementation of GUI in devices that started in the early 1990s has now also spread to embedded systems [29,30,31,32,33]. This innovation led to the increase in research and funding towards improving the user interface instead of the performance enhancement of the equipment. Especially, a strong hardware and software platform and numerous programming personnel are required to maintain the performance of the GUI as well as meet the demand of the users. To address this issue, many embedded systems divide the system into a host-local structure, using standard computing devices such as a common personal computer (PC) as the host. With the rapid advancement in smart devices since the 2000s, it is speculated that the host will soon be smart devices, and the standard link between the host and local systems will be achieved through wireless internet (WiFi) or Bluetooth.

Current biomedical equipment can be connected to the cloud to construct a big data platform, in which a top-level application will extract the data to be uploaded to the cloud. However, this structure has limitations in simultaneous monitoring and control between the data and the equipment itself as seen in ‘industrial internet of things.’ Considering the product lifetime management, employing a cloud server for equipment maintenance, aging analysis, and user interaction will be much simpler if the abstraction layer from the hardware to the end software is in a distributed structure.

This paper proposes a cloud-based software architecture for an automated POC-qPCR device considering all the factors aforementioned that emerged from the advancement in technology. The distributed structure was selected instead of the top-down structure that is observed in current equipment, and the communication between distributed functions was achieved using a network to allow access to detailed functions via the cloud. In addition, the representation state transfer (REST) application programming interface (API) server was implemented to allow web-based GUI access. The nucleic acid diagnosis process was divided into DNA extraction and qPCR, which can be controlled and monitored separately via the cloud. The qPCR control and monitoring includes both the PCR amplification and fluorescence detection as in commercial qPCR equipment. An emulator was manufactured to evaluate the applicability of the proposed software architecture. Functional verification was performed through individual and simultaneous access by implementing not only a web-based GUI that performs full automatic molecular diagnosis, but also a GUI that monitors and controls extraction and that for qPCR functions. The experimental results confirmed that the software architecture proposed in this paper can be used to successfully control the entire molecular diagnostic process including DNA extraction and qPCR, as well as to simultaneously monitor the operation in real time.

## 2. Materials and Methods

### 2.1. Target Hardware Architecture

#### 2.1.1. Microfluidic Cartridge

The cloud-based software architecture was built to automate DNA extraction and qPCR using the cartridge shown in Figure 1 (LabGenius^TM^ Cartridge, Biomedux, Suwon, Korea). The cartridge consists of an extraction body that extracts DNA with magnetic beads, and a PCR chip where amplification and fluorescence detection occur (Figure 1a). It has already been shown in many pieces of research that magnetic bead DNA extraction results in higher concentration and less contamination in the extracted DNA compared to the method using silica membranes [34]. The cartridge has several isolated chambers that contains reagents necessary for each step of magnetic bead DNA extraction, which involves cell lysis, DNA adsorption, washing, and DNA desorption (Figure 1b). Once the patient sample is loaded in the sample chamber, the sample will be transported to each chamber by the rotating valve at the bottom of the cartridge and the syringe in the middle for DNA extraction (Figure 1c). For full automation of this cartridge, control and operation of the syringe stepper motor, rotation valve stepper motor, and the magnet position servo motor is required.

The extracted DNA is then transported to the PCR reaction chamber of the PCR chip through the microchannels between the two bodies. The structure of the PCR chip has been previously reported [35]. Briefly, the reaction chamber is made of polycarbonate and attached to the PCB substrate with a heater pattern and thermistor (Figure 1d). For a successful amplification, temperature monitoring via the thermistor and operating the heater pattern and fan during thermal cycle is crucial.

As the PCR cycle starts, the fluorescence is detected concurrently from the transparent side of the PCR chip as can be seen in Figure 1a. The fluorescence detection system consists of four LEDs and excitation filters closely placed at 45° to the reaction chamber, and an objective lens, an emission filter wheel, and an ocular lens that is aligned to direct the fluorescence to a photodiode. In fluorescence detection, the software needs to be able to operate the filter wheel to correctly position the emission filter, turn on the LED, and read the photodiode in sequential order.

#### 2.1.2. Driving System for the Cartridge

The overall system structure of the target hardware is illustrated in Figure 2. A Raspberry Pi computer was selected for the single board computer (SBC), which is the main controller in this system, for ease of maintenance and versatility. In particular, the Pi3A+ model was chosen considering the size, power, processing speed, and performance. A separate microcontroller (PIC18F4553, Microchip technology Inc., Chandler, AZ, USA) was incorporated to control the temperature of the PCR chip since frequent control of 2 ms is required.

The motors involved in DNA extraction and the emission filter wheel for fluorescence detection was controlled directly by the SBC. In detail, the syringe, rotation valve, and the filter wheel stepper motor were controlled by a motion controller (L6470, STMicroelectronics, Geneva, Switzerland) wired to the SBC with an Inter-Integrated Circuit (I2C) interface. Because this motion controller only provides a Serial Peripheral Interface (SPI), the protocol was converted using an I2C-bus to SPI bridge chip (SC18IS602B, NXP Semiconductors, Eindhoven, The Netherlands). By doing so, the number of wires required can be reduced from four (SPI) to two (I2C). The magnet servo motor was wired directly to the SBC since it is controlled only with a pulse width modulation (PWM) signal.

The cooling and heating of the PCR chip was achieved by controlling the fan and heater pattern on the PCB substrate with PWM through the general purpose input output (GPIO) port of the microcontroller (Figure 3). The selected microcontroller is able to measure the resistance of the thermistor and also read the photodiode output since it includes a 12 bit analog to digital converter (ADC). Turning the excitation LEDs on and off was also controlled by this microcontroller to maintain consistency between fluorescence detection and excitation. The microcontroller measures the thermistor’s resistance and converts it to temperature every 2 ms as commanded by the SBC, then performs proportional–integral–derivative (PID) temperature control using the heater pattern and a fan. When the SBC commands the return of the photodiode value at a set fluorescence filter, the microcontroller turns on the respective LED and reads the photodiode value to return. The SBC and microcontroller is connected by a universal serial bus (USB), exchanging information every 50 ms using a 64 byte IN/OUT packet.

### 2.2. Software Architecture

Figure 4 shows the block diagram of the proposed software architecture. Operation of hardware related to DNA extraction including syringe and rotation valve movement and magnet positioning is managed by the extractor controller, whereas hardware related to qPCR is managed by the PCR controller. Both modules were constructed as a zero message queue (ZMQ) socket server, allowing the main server to access from any nodes in the cloud hierarchy. In this study, the main server is in the SBC, which can be regarded as the terminal device in the cloud hierarchy. The protocol manager creates, reads, updates, and deletes the extraction and PCR protocols in the SQLite database (DB) according to the user interface. The extraction and PCR interface reads and executes the protocol sent by the protocol manager line by line. The hardware controllers and the protocol manager were established with the REST-API as shown in the ‘main server’ block in Figure 4.

The Python Flask web server was used for the user interface, and the front-end was divided with a React framework. The web server receives the user request and controls the protocol manager, and the PCR and extractor interfaces through REST-API. The extractor interface requests a service to the extractor controller, which operates the magnet servo, rotation valve motor, and syringe motor. Similarly, the PCR interface will request a service to the PCR controller, which then controls the microcontroller and L6470 connected to the PCR chip and filter wheel, respectively.

### 2.3. Operation Example

The interaction between the modules during DNA extraction and PCR detection is elucidated in this section. Table 1 displays the set of the high-level commands by means of which the main server and extractor controller communicate during DNA extraction, where each command can have a maximum of two parameters. The following explains only the commands related to the operation example shown in Figure 5. The ‘home’ command moves the rotation valve and syringe to the home position. The ‘goto’ command receives the first parameter, the integer ‘n’, and transports the sample to the ‘n’th chamber of the extraction body. For the ‘pumping’ command, the first parameter can be ‘sup’, ‘sdown’, ‘up’, or ‘down’, which commands the syringe to slowly go up, slowly go down, go up at normal speed, and go down at normal speed, respectively. Stopping the ‘pumping’ command is determined by the second parameter, where if it is an integer ‘n’ it commands to stop at n milliliters, and ‘full’ commands to pull up or push down the syringe all the way.

Figure 5 shows an example of a simple interaction scenario. The extractor interface reads the first line of the extraction protocol ‘home’ and transmits the command to the extractor controller. The extractor interface will only proceed to the next line after the command ‘home’ has been completely processed. Since the second line starts with ‘%’, the extractor interface will skip to the third line ‘goto 5’. The process is repeated line by line until the end of the protocol.

Next, we observe the interaction between modules during qPCR. Each row of the PCR protocol shown in Table 2 can be written as a unit action A(i), represented by a 3-component tuple of (Label, Temperature, Duration). Then, the protocol can be regarded as the sequence of the unit action Q as follows;
(1)Q={A(i)|A(i)=(L,T,D),i=1~n}
where n is the number of the actions, that is, the rows. The unit action label ‘SHOT’ commands to measure the fluorescence with the photodiode, and ‘GOTO’ commands to jump D times to a unit action with the label T. Defining the unit action as such allows a uniform equation for the sequence Q, as in Equation (1). The component L for each unit action can be a positive integer, ‘GOTO’, or ‘SHOT’. T can either be a temperature or a label, and D can be time in seconds or quantity of jumps. The PCR interface executes the sequence Q as shown in the Algorithm 1 ‘RealTimePCRproc’ procedure.
**Algorithm 1** RealTimePCRproc.1.*n* = no. of lines in the protocol2.*i* = 13. do while *i <= n*4.    fetch *A*(*i*) 5.    if *A*(*i*)*.L* == SHOT6.       wait shot ()   % read photodiode7.    else if *A*(*i*)*.L* == GOTO8.       if *A*(*i*)*.D* != 09.           *i = A*(*i*)*.T*  % label where jump to10.         *A*(*i*)*.T--*   % no. of jumps11.       else12.         *i++*13.       endif14.    else15.       Send target temperature *Tc* to PCR controller and wait until *|Tc-A*(*i*)*.T| <*
*ε*16.       wait SecTimer(*A*(*i*)*.D*) == 017.    endif

The sixth line of the procedure depicts what happens when the first ‘Label’ component is ‘SHOT’, where the Shot() function returns the fluorescence intensity through the photodiode. Lines 8 to 13 instructs the operation when the ‘Label’ component is ‘GOTO’. Here, ‘Duration’ is the number of jumps commanding a jump to *A*(*i*)*.T* unless the value is ‘0’. Unit action *A*(*i*)*.D* decreases by 1 every jump, and the operation moves on to the next unit action once this values reaches zero. Lines 15 and 16 orders what happens when ‘Label’ component is neither ‘GOTO’ nor ‘SHOT.’ The 15th line instructs to wait until the temperature difference between the target (*Tc*) and chip temperature is less than the predefined threshold *ε* °C. The ‘SecTimer(*A*(*i*)*.D*)’ function at the 16th line initializes the timer as *A*(*i*)*.D* seconds and returns the decremented time every second.

Figure 6 illustrates the interaction scenario in which the PCR interface, the PCR controller, and the web GUI module communicate every 100 ms. The PCR interface requests the status of the PCR controller and stores the information every 100 ms. The web GUI sends an http request to the PCR interface every 100 ms independently, and updates the web GUI with the status received from the PCR controller.

The interaction scenario during the whole molecular diagnosis, which includes DNA extraction, PCR, and the detection, is shown in Figure 7 sequentially. The extraction protocol starts by the user clicking the start button at the web GUI, which will send a request to the main server by REST-API. When the main server receives the request, it will read the stored protocol and start communicating with the controllers. The main server and extractor controller communicate every 100 ms until the extraction protocol is done, at which point the main server will start communicating with the PCR controller every 100 ms. During this process, the main server will send the ‘Run’ command and unit action of the PCR protocol sequentially to the PCR controller until PCR is complete.

In this software architecture, it is possible to pause/stop while the protocol is running even though it is not shown in Figure 7. The user can request to stop the operation through the main GUI, which will transmit the request to the main server through the stop API. When the main server receives this request, it will send the ‘stop’ command to whichever controller it was communicating with at the moment.

Figure 8 shows the protocol manager interaction scenario. When the web GUI requests a list from the protocol manager, it will read the protocol list from the database and upload to the web GUI. To create a new protocol, the user requests a ‘new’ request through the REST-API from the web GUI and sends the new protocol data. The new protocol data received is then saved at the SQLite DB by the protocol manager. Updating and deleting an existing protocol is also requested through the REST-API, where both the protocol data and index is sent to the protocol manager. It will locate the protocol with the same index from the SQLite DB and either update or delete the protocol according to the request sent.

### 2.4. Emulator

An emulator was constructed to evaluate the performance of the proposed software architecture (Figure 9). The ‘SBC module’ at the left bottom of the figure shows the Raspberry Pi 3A+ and the hat board with power and I2C connectors. The ‘motion controller module’ is linked to the SBC hat board with an I2C cable consisting of three pairs of an I2C-bus to SPI bridge chip and a motion controller, where one of the motion controllers is connected to a stepper motor that emulates all of the syringe, rotation valve, and filter wheel motor. An RC servo motor was used as the ‘servo motor’ in this emulator and operates by accessing the magnet servo. To emulate the PCR and detection operation shown in Figure 3, ICP12-USB stick (iCircuit Technologies, www.piccircuit.com) denoted as ‘microcontroller module’ was selected. This has the PIC18F2553 (Microchip technology Inc., Chandler, AZ, USA), which only differs in the number of GPIO ports with the microcontroller employed in the target system. The firmware is exactly the same as that of the microcontroller of the target system, but is programmed to compile as an emulator mode and generates the photodiode and temperature to be sent to the PCR controller, disregarding the ADC.

### 2.5. Validation of the Proposed System Architecture

Three types of web GUIs were established to operate the emulator to demonstrate the validity of the proposed software architecture. A web GUI was constructed to evaluate the whole molecular diagnostic process with the protocol for detection of *Chlamydia trachomatis* (CT) and *Neisseria gonorrhoeae* (NG). We made separate web GUIs for the extractor controller and the PCR controller to make sure there were no problems with cloud control and monitoring.

## 3. Results

### 3.1. Whole Molecular Diagnostic Process

To validate the proposed software architecture for the whole molecular diagnostic process, a three-page web GUI was made using React library and Bootstrap framework. Clicking the ‘setup’ button on the main page that manages the overall protocol will redirect the page to the setup page. From there, users can go to the editor page by clicking the ‘add’ or ‘edit’ button.

The main page consists of six groups with bootstrap components and one plot (Figure 10). The ‘Connection’ group includes html label elements for the serial number of the target hardware and the connection status. The ‘Progress’ group contains the name of the protocol, status of operation, and remaining time until the protocol is complete. The ‘Device’ group has a label element to provide the current temperature of the PCR chip. The ‘Protocol’ group includes a select element with the list of stored protocols, a button element to start or stop the operation, and another button element to go to the setup page. The ‘Cq Value’ group has the image button element that allows the users to select which fluorescence value to be shown in on the plot, and a label element that displays the cycle quantification value (Cq) after PCR is complete. Finally, the ‘Result’ group consists of a table element showing the fluorescence name and positive/negative result of the PCR. The plot is shown as a 2D plot, where the *x*-axis of the plot represents the PCR cycle, and the *y*-axis represents the fluorescence intensity received from the PCR controller.

At the setup page, the users will see the ‘Protocol Manager’ and ‘History’ group. The ‘Protocol Manager’ consists of the data list element that shows the list of protocols, and a button element ‘Add’, ‘Edit’, and ‘Delete’ to update the protocols. The ‘History’ group is shown in a table element that lists the records of protocols that were run previously.

The editor page has the ‘Filter (Label, CT)’, ‘PCR Protocol’, and ‘Extractor Protocol’ text areas, ‘Protocol name’ input element to enter the protocol name, ‘Save’ and ‘Cancel’ button elements. ‘Filter (Label, CT)’ group has input elements to enter the target DNA names and set the cycle thresholds for detecting Cq and image button elements to select the fluorescence dyes to use is displayed. Both the PCR protocol and the extraction protocol are entered into ‘PCR Protocol’ and ‘Extraction Protocol’ text areas, respectively. ‘Save’ button element is for saving the protocol with the name in ‘Protocol name’ and data in the ‘Filter (Label, CT)’, ’PCR Protocol’, and ‘Extraction Protocol’, and returns the user to the setup page. The ‘Cancel’ button element returns to the setup page without saving any of the data that was input.

Using the web GUI and emulator described above, the protocol for multiplex detection of CT and NG is tested. Precise operation of the stepper motor was observed in the DNA extraction step, and accurate temperature and fluorescence readout from the microcontroller was displayed during PCR. The 2D plot and the positive/negative table of the result group shows the emulated diagnostic results. The 2D plot was for the emulated RFU’s stored in the microcontroller module. As both RFU’s emulate the positive results, the results for both of CT and NG were positive, as shown in Figure 10b.

### 3.2. Extractor Controller Monitoring

The extractor monitoring web GUI to evaluate the extractor controller was constructed using a Vue.js framework to have the ‘Protocol Command’ group to operate DNA extraction, ‘Register’ group that can observe and set the registers of the motion controller, and ‘Motion test’ group to control the motor. Commands ‘home’, ‘goto’, ‘waiting’, and ‘pumping’ can be done in the ‘Protocol Command’ group. The ‘Register’ group consists of input elements to set the register value of the motion controller (L6470) to control the position, speed, acceleration, and current of the motor. The ‘Motion test’ group has radio button elements allowing the user to select the motor to be tested between the rotation valve and syringe motor, and button elements to test basic stepper motor motions such as ‘jog’, ‘home’, ‘go until’, and ‘move’.

The operation of the extractor controller was investigated using the functions of the extractor monitoring web GUI for the emulator. The experimental results verified the normal operation of the extractor controller over the internet, evidencing the cloud-based control of the extractor controller.

### 3.3. PCR Controller Monitoring

Python and a Jupyter notebook were utilized to demonstrate the PCR controller controllability over the internet. Accurate temperature and remaining time were displayed by the PCR controller while running the example PCR protocol with four color multiplex. Precise movement of the stepper motor to the four respective filters and back to the home position was observed while running ‘SHOT’ command in the protocol. Furthermore, successful monitoring of the protocol progress including current chip temperature, fluorescence intensity, cycle number, and remaining time was achieved when running the molecular diagnostic protocol. The experimental results verified the normal operation of the PCR controller over the internet.

## 4. Discussion

This paper proposes a cloud-based software architecture for a fully automated POC molecular diagnostic system. The two fundamental steps of molecular diagnosis, DNA extraction and qPCR, were stablished as socket servers to allow easy access via cloud. The management and operation of the protocol was executed with a REST-API server that can be accessed at any node of the cloud hierarchy covering the terminal device, edge, and cloud. Since this server also includes the user interface, users can access the system through a web-based UI on any smart computing device. Note that the web-based user interfaces are very helpful for close collaboration between UX designers and software developers because basic functional development and design development are separated.

The proposed software architecture holds high potential of being integrated into various biomedical instruments since it can be used to perform any physical sensing such as temperature and fluorescence, and also manages controls that standard actuators can do such as controlling the position, temperature, and fluidic quantity. Furthermore, additional modification can lead to the implementation of this cloud-based software architecture to various cyber-physical systems in other fields. The architecture proposed in this paper will be ported to the target system currently under development, and it is expected that it will be more refined and verified in this porting stage.

Raspberry Pi was determined to be the main controller system among the many open platforms considering the user population. However, once the equipment is integrated to the cloud and only functions for sensors and actuators, a microcontroller can replace the Raspberry Pi (SBC). In addition, given that the proposed software architecture is written with Python, we speculate that it can easily be employed to any equipment with a rapidly advancing WiFi microcontrollers programmable in Python.

## Figures and Tables

**Figure 1 sensors-21-06980-f001:**
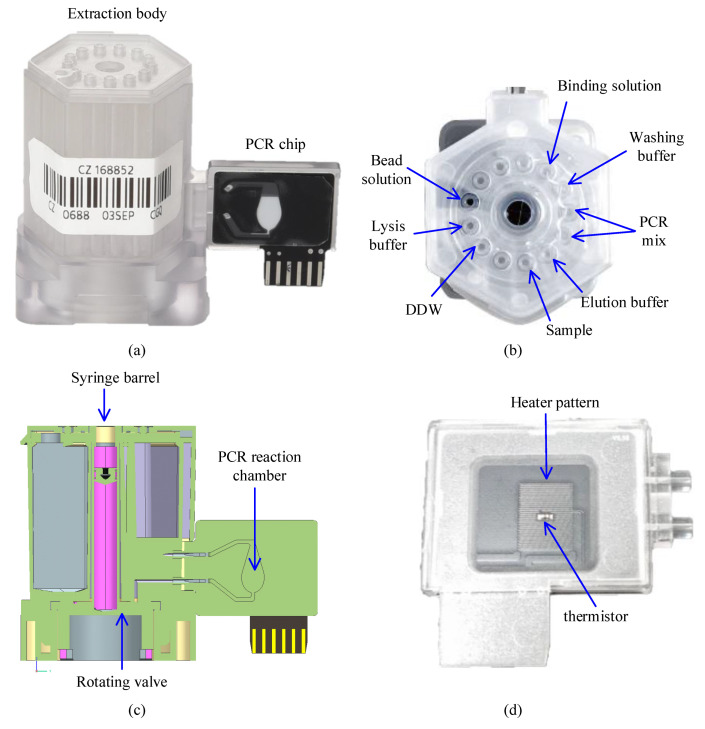
The actual cartridge used to demonstrate the performance of control with the proposed software. (**a**) Side view of the cartridge; (**b**) Top view of the extraction body showing an example of reagents allocation into each chamber; (**c**) Cross section illustration of the cartridge; (**d**) The back-side picture of the PCR chip which shows the heater pattern and thermistor attached to the PCB.

**Figure 2 sensors-21-06980-f002:**
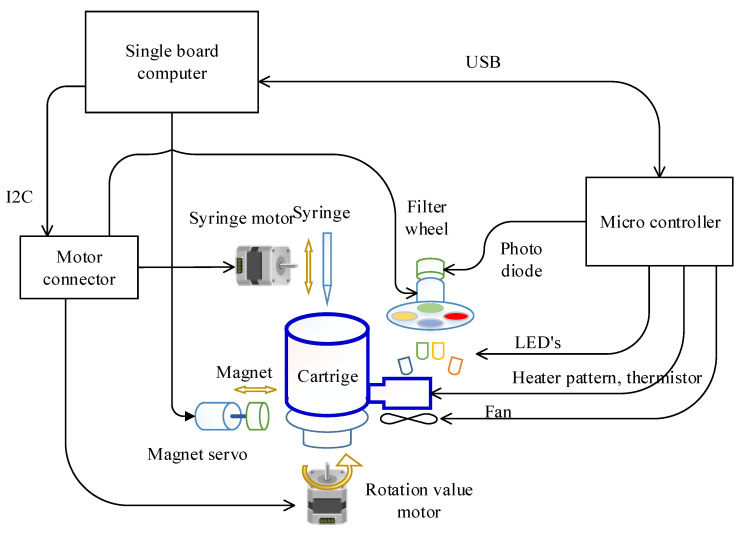
Schematic of the overall target system.

**Figure 3 sensors-21-06980-f003:**
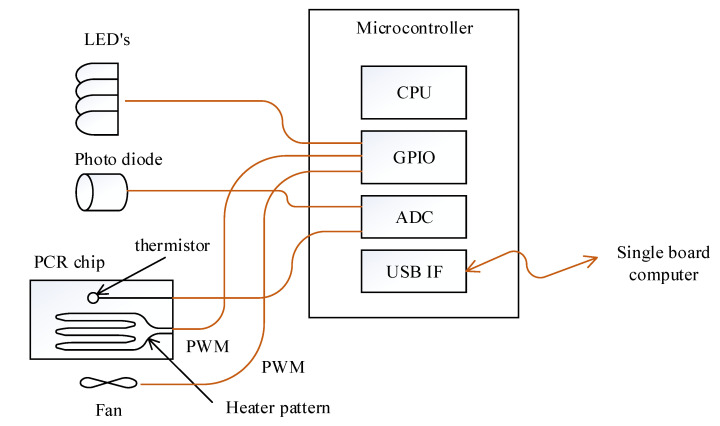
Schematic of the microcontroller and its peripheral. The connection between the microcontroller and excitation LEDs and PCR chip is also illustrated.

**Figure 4 sensors-21-06980-f004:**
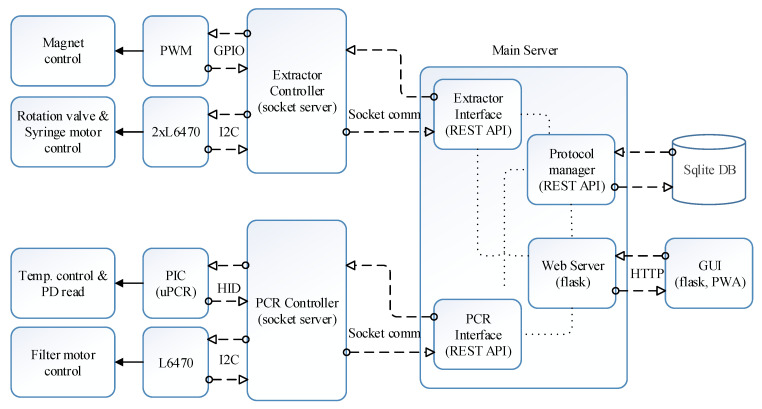
Software architecture block diagram.

**Figure 5 sensors-21-06980-f005:**
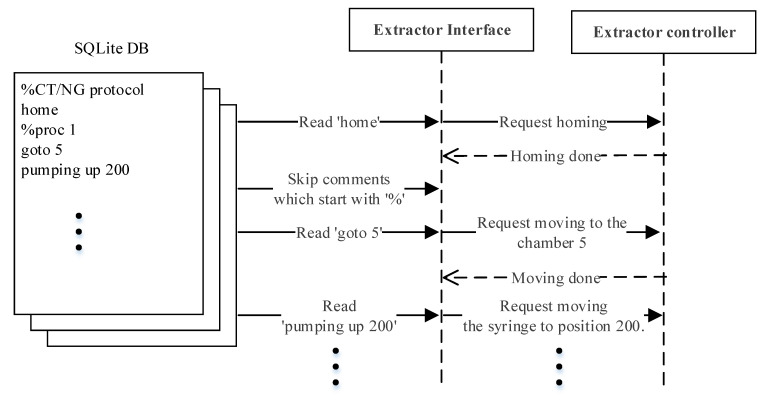
DNA extraction command flow example.

**Figure 6 sensors-21-06980-f006:**
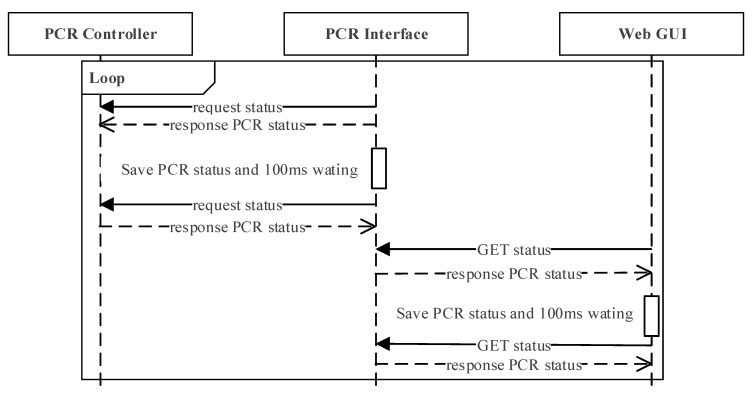
Interaction scenario in which PCR interface, PCR controller, and web GUI module communicates every 100 ms.

**Figure 7 sensors-21-06980-f007:**
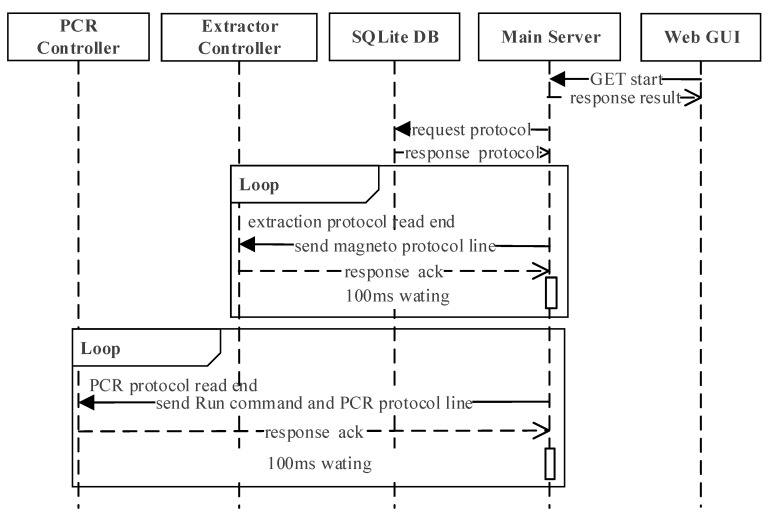
The interaction scenario during the whole molecular diagnosis in sequential order.

**Figure 8 sensors-21-06980-f008:**
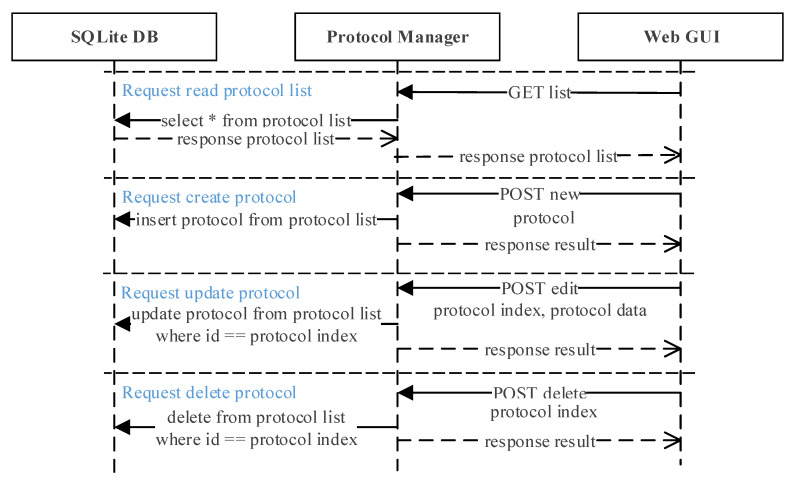
Protocol manager interaction scenario. Protocols can be read, created, updated, and deleted by sending a simple request from the web GUI to the protocol manager.

**Figure 9 sensors-21-06980-f009:**
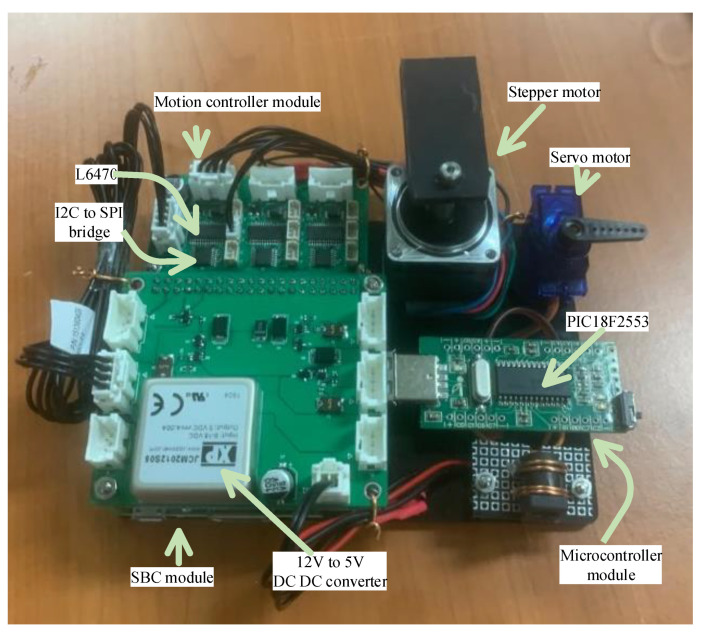
Emulator to evaluate the performance of the proposed software architecture.

**Figure 10 sensors-21-06980-f010:**
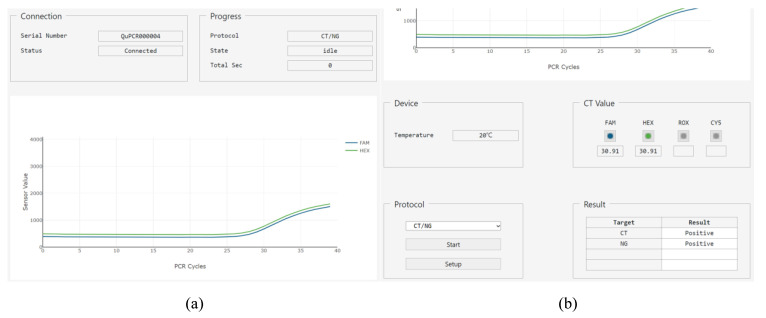
Main page when the protocol has finished. (**a**) The upper part of main page, which shows two bootstrap components and the plot; (**b**) The lower part of main page with the other four bootstrap components.

**Table 1 sensors-21-06980-t001:** High-level command set for DNA extraction.

Command	Param1	Param2
home		
waiting	n	
goto	n	
pumping	sup/sdown/up/down	n or full
magnet	on/off	
getStatus		

**Table 2 sensors-21-06980-t002:** PCR protocol example.

Label	Temperature (°C)	Duration (s)
1	95	30
2	95	30
3	55	30
4	72	30
SHOT		
GOTO	2	39
5	72	180

## Data Availability

Not applicable.

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
