# Peer review of "Cloud-Based Software Architecture for Fully Automated Point-of-Care Molecular Diagnostic Device"

_sensors, 2021, doi:10.3390/s21216980_

Round 1

Reviewer 1 Report

This paper summarized research progress of cloud-based software architecture for full automated point-of-care molecular analysis devices. This provides a detailed systematic and section-wise overview of equipment. The authors of this manuscript understandably present this work and show a complete understanding of the subject. However, I have some suggestions for authors to help improve the review manuscript.

  1. The schematic diagram of the equipment is not clear enough to understand by the reader. Like only one chamber is shown here which is used to take the sample. But for sample washing adsorption and desorption no chamber is defined here.
  2. As the reaction chamber is made with polycarbonate and attached to the PCB substrate but no information present about the PCB Substrate.
  3. The second opinion is introduction and Material of equipment topics both are too comprehensive. It is good to describe each and everything but it should be compact and concise so the reader should not be confused instead of leaning this.
  4. Table 1 and Figure 5 are really hard to follow, the authors should better redraw these for easier understanding.
  5. As for Figure 9, the authors only point out the names of few parts, it is suggested to describe all necessary parts.
  6. Regarding the test performance, the authors should present the detailed results rather than general descriptions, such as “Precise operation of the stepper motor was observed in the DNA extraction step, and accurate temperature and fluorescence readout from the microcontroller was displayed during PCR”

Author Response

Dear Reviewer,

Thank you very much for the comments for the following paper:

Ms. Ref. No.: sensors-1397942

Manuscript Title: Cloud-based software architecture for fully automated point-of-care molecular diagnostic device

The responses on your comments are in the attached file and start with ‘ >> ’ after each comments.

Kind regards,

Jong Dae Kim

Reviewer 2 Report

See the attachment for my feedback.

Author Response

(The authors gave the same response as above.)
